# Phosphasilazanes as Inhibitors for Respirable Fiber Fragments Formed during Burning of Carbon-Fiber-Reinforced Epoxy Resins

**DOI:** 10.3390/molecules28041804

**Published:** 2023-02-14

**Authors:** Philipp Kukla, Lara Greiner, Sebastian Eibl, Manfred Döring, Frank Schönberger

**Affiliations:** 1Fraunhofer Institute for Structural Durability and System Reliability LBF, Schlossgartenstr. 6, 64289 Darmstadt, Germany; 2Bundeswehr Research Institute for Materials, Fuels and Lubricants, Institutsweg 1, 85435 Erding, Germany

**Keywords:** epoxy resin, phosphasilazanes, carbon fiber, fiber degradation, cone calorimeter

## Abstract

Carbon-fiber-reinforced polymer composites (CFRPs) exhibit additional hazards during and after burning due to respirable fragments of thermo-oxidatively decomposed carbon fibers. In this study, various phosphasilazanes are incorporated into the RTM 6 epoxy matrix of a CFRP to investigate their flame-retarding and fiber-protective properties via cone calorimetry. Residual carbon fibers are analyzed using SEM and EDX regarding their diameter and elemental composition of deposits. The decomposition process of phosphasilazanes is characterized by DIP-MS and infrared spectroscopy of char. Flame-retardant efficiency and mode of action are correlated with the chemical structure of the individual phosphasilazane and compared for neat resin and composite samples. Phosphasilazanes mainly acting in the condensed phase show beneficial fiber-protective and flame-retardant properties. Those with additional gas phase activity are less efficient. The phosphasilazanes degrade thermally via scission of the Si-N bond. The distribution and agglomeration of deposited particles, formed during the fire, influence the residual fiber diameters. Continuous layers show the best combination of flame retardancy and fiber protection, as observed for *N*-dimethylvinylsilyl-amidophosphorus diphenylester.

## 1. Introduction

Carbon-fiber-reinforced polymer (CFRP) composites show beneficial properties compared to pure plastics since they combine properties of the single materials in the lightweight composites. Epoxy resins are often used as high-performance matrices, especially in sport goods, transportation, or aviation due to their thermal and environmental resistance and insolubility [1,2,3,4]. Depending on the used fibers, electrical, thermal, and mechanical properties can be adjusted [1,5,6].

Thermo-oxidative degradation of the carbon fibers (CFs) in such composites represents an additional hazard in burning events in addition to heat and toxic gases. In an accident with an external heating source, the fibers are decomposed and detectable in the generated soot [7,8,9]. Because fibers with diameters lower than 3 µm, a length greater than 5 µm, and an aspect ratio greater than 3:1 are respirable, they are considered cancerogenic by the World Health Organization (WHO) [10]. Respirable fibers were detected in the residue of irradiated samples as well as on the protective suit of firemen in a large-scale burning test. The surface of these fiber fragments is porous and can, thus, take up toxic combustion products. Hence, fiber protection is necessary for burning events in order to inhibit fiber decomposition [11,12,13,14,15].

Fiber protection can be achieved via protective layers, such as char, glass, or intumescing barriers [16,17]. Glass or ceramic structures can be applied as a coating on fibers. Protective coatings on composites prohibit the exchange of heat and combustible volatiles. Therefore, matrix degradation is hindered and, thus, they act as flame retardants. A combination of char and glass formation acts synergistically. The yield of residues is higher than expected. Specifically, a combination of silicon and phosphorus-containing compounds is beneficial. The silicon moieties form a polar glass that cannot bind to nonpolar char or carbon fibers, but to glass fibers. On the other hand, the phosphorus moiety is able to form phosphates that bind covalently to the char [18,19,20]. Additionally, it promotes char generation [18,19,20].

These coatings can be generated from pre-ceramic polymers in an extra processing step [21,22,23]. Polysilazanes, for example, generate ceramics, such as silicon carbide (SiC), silicon nitride (Si_3_N_4_), and hybrid materials thereof (SiCN). [24] These ceramics can be incorporated into carbon fibers. Therefore, they enhance the tensile strength of such composites, especially at elevated temperatures of up to 400 °C [21,25].

The abovementioned ceramic coatings have a negative impact on overall mechanical performance, weight, or price. For example, the interlaminar shear strength is decreased by 25% by such coatings, leading to insufficient properties for applications in aviation and aerospace [26]. Hence, a different approach is used in this paper. Instead of a ceramic coating, the pre-ceramic structure is incorporated into an epoxy resin matrix. The pre-ceramic structure generates a protective layer during the burning event. The interlaminar shear strength is less decreased by the incorporation of flame retardants into the matrix and their content can be varied.

Silicones and siloxanes show good in situ flame-retardant properties regarding the matrix in carbon-fiber-reinforced composites [27,28]. They also result in negative effects, especially in promoting fiber degradation [29]. On the other hand, phosphorus-containing moieties acting in the condensed phase show both flame-retardant as well as fiber-protective properties [29,30]. Combinations of phosphorus and silazane moieties provide promising results for glass-fiber-reinforced composites [31,32,33]. Therefore, this paper focuses on phosphasilazanes for carbon fiber protection.

In our previous work, we developed novel phosphasilazanes, which contain both phosphorus and silazane moieties [34]. Flame-retardant properties were achieved in neat epoxy resin depending on the chemical environment at the phosphorus atom. Phosphasilazanes with high oxygen content promoted char generation or even intumescence in the case of the structure containing salicylic acid. This led to higher char yields and better flame-retardant performance in both thermo-analysis and cone calorimetry. On the other hand, a higher amount of silazane groups within one molecule at the same concentration increased flammability. Phosphorus moieties acting in the gaseous phase were antagonized with the generation of char [34].

In this paper, the potential of these phosphasilazanes is examined to inhibit the generation of respirable carbon fiber fragments via cone calorimetry. The residual fibers and char were analyzed using SEM, EDX, and IR spectroscopy. Additionally, the mode of action of this type of flame retardant in carbon-fiber-reinforced epoxy resin is described. Mass spectroscopy was applied to reveal the degradation modes of phosphasilazanes. This way, knowledge on the mechanisms of flame retardancy and inhibition of fiber degradation is attained.

## 2. Results

### 2.1. Burning Behavior—Cone Calorimetry

Flame-retardant effects are characterized in comparison to neat resin samples from our previous publication (Figure 1) [34]. We showed that the chemical environment of the phosphorus atom in the phosphasilazane affects fire properties in the neat epoxy resin. Specifically, phosphasilazanes with an oxygen-rich environment at the phosphorus atom leading to a mode of action in the condensed phase showed improved flame retardancy in neat resin. For PO(OPh)_2_-VSil **1**, charring occurs. SCP-VSil **3** also exhibits intumescing char. Both structures contain two OR moieties at the phosphorus atom. On the other hand, structures with increased silazane content (POOPh-VSil_2_ **2** and Spiro-VSil_2_ **5**) or with a predominant mode of action in the gaseous phase such as DOPO-VSil **4** provide less flame retardancy due to a carbon- or nitrogen-rich environment at the phosphorus atom. The vinyl moiety does not affect the mechanical properties of the matrix. It is degraded at lower temperatures than a methyl moiety [35]. This leads to the methyl moiety degrading after the matrix and, thus, to less effective flame-retardant properties than in the presented vinyl moieties.

In this study, cone calorimetry is used for the characterization of carbon-fiber-reinforced samples with phosphasilazanes (Figure 2). In the RTM 6 composite without any phosphasilazane **0**, the heat release rate (HRR) rises fast after ignition, leading to multiple peaks originating from delamination processes [12,36,37]. During delamination, adjacent carbon fiber layers are separated. Hence, additional volatiles are released abruptly. Their combustion leads to a higher HRR. Afterwards, the HRR is decreased until the next delamination. For the CFRP containing phosphasilazanes, no sharp HRR peaks are observed. This is analogous to other silazane-containing carbon-fiber composites, explained with a slower, more continuous burning of the sample [38,39]. A single and comparably small peak is observed for PO(OPh)_2_-VSil **1** and POOPh-VSil_2_ **2**. Two peaks with a higher intensity arise for SCP-VSil **3**, DOPO-VSil **4,** and for Spiro-VSil_2_ **5,** indicating a charring effect for these samples. The second peak describes less-efficient barrier effects compared to PO(OPh)_2_-VSil **1** and POOPh-VSil_2_ **2** by the formed char at the end of combustion, when the pyrolysis front reaches the back of the sample. The char might even degrade, leading to worse flame-retardant properties for SCP-VSil **3**, DOPO-VSil **4,** and for Spiro-VSil_2_ **5**.

The pHRR is reduced for all samples containing phosphasilazane flame retardants (Table 1). PO(OPh)_2_-VSil **1** shows the highest reduction of 52%, down to 260 kW m^−2^ and, thus, is considered the most effective flame retardant from this series. POOPh-VSil_2_ **2** and DOPO-VSil **4** also decrease the pHRR by around 30%. The least effective flame retardants are SCP-VSil **3** and Spiro-VSil_2_ **5,** which still reduce this parameter by 12%. The same trend is observed for the maximum average rate of heat emitted (MARHE).

In order to compare the total heat release (THR), it must be normalized to the matrix content (X), whereby X describes the mass ratio of resin and flame retardant to the overall sample mass. The obtained values for X vary between 0.27 and 0.49, which is typical for hand lamination processing [40,41]. The THR X^−1^ is equal to the reference sample of RTM 6 **0** for SCP-VSil **3** and Spiro-VSil_2_ **5** and, thus, these samples do not indicate flame retardancy. The parameter is reduced for PO(OPh)_2_-VSil **1** and POOPh-VSil_2_ **2** by over 20% to 55 MJ m^−2^ and 57 MJ m^−2^, respectively.

The total smoke release per matrix content (TSR X^−1^) also hints at the mode of action. If this value is decreased, it describes a mode of action in the condensed phase, whereas an increased TSR X^−1^ indicates a mode of action in the gaseous phase, like for THR X^−1^. This parameter is comparable for SCP-VSil **3** (3820 m^2^ m^−2^) to the carbon-fiber-reinforced resin **0** (3760 m^2^ m^−2^). Hence, the TSR X^−1^ also does not indicate an efficient flame retardancy for SCP-VSil **3**. The other phosphasilazanes and especially PO(OPh)_2_-VSil **1** as well as POOPh-VSil_2_ **2** decrease this parameter by up to 45% (2290 m^2^ m^−2^ and 2130 m^2^ m^−2^). Therefore, it supports the result of action in the condensed phase.

The residual mass (m_R_) is normally another indicator for the mode of action, but this value is not significant due to variation in matrix content in the samples resulting from the hand lamination process.

The time to ignition (TTI) is reduced by 5–13 s for the various phosphasilazane-containing samples, except for SCP-VSil **3** compared to the pure RTM 6 composite. The reduction in TTI is common for flame retardants acting in the condensed phase since the surface heats faster. Because of this and the double peak in HRR, PO(OPh)_2_-VSil **1,** POOPh-VSil_2_ **2**, DOPO-VSil **4,** and Spiro-VSil_2_ **5** are considered to act partially in the condensed phase in the CFRP samples. On the other hand, the pHRR, MARHE, and THR X^−1^ are not influenced for Spiro-VSil_2_ **5**. Therefore, this molecule shows neither flame-retardant nor flame-accelerant properties. The time to flameout (TTF) is not influenced significantly by the phosphasilazanes compared to RTM 6 **0**.

These results suggest two different modes of action for the flame-retardant properties in the phosphasilazanes. On the one hand, a reduced TTI in combination with decreased pHRR and THR X^−1^ describe increased flame retardancy in the condensed phase [42,43], as shown before for PO(OPh)_2_-VSil **1** in neat resin [34]. On the other hand, reduced times to ignition with a relatively high pHRR indicate fuel-like behavior, leading to worse flame-retardant performance, like for Spiro-VSil_2_ **5** [44]. This would be expected from the silazane moiety because of its flammability [45]. Therefore, in principle, two competing effects occur in carbon-fiber-reinforced samples, depending on the phosphasilazane.

### 2.2. Burning Behavior—UL94 Burning Test

The results obtained by cone calorimetry indicate differences in flame-retardant properties for the various phosphasilazanes, which is supported by vertical UL94 burning tests.

Despite some samples having a V-1 or V-0 classification in neat resin formulation [34], the investigated composite samples achieve no classification (n.c.) in vertical UL94 burning tests (Table 2). This may result from carbon fibers weakening the flame-retardant effect in the condensed phase by increasing the sample surface (see below). Still, a decrease in the mean first burning time t_1_ is detectable by the addition of phosphasilazanes. Whilst PO(OPh)_2_-VSil **1** has a high mean first burning time t_1_ of 44 s, it is below 10 s for SCP-VSil **3**, DOPO-VSil **4**, and Spiro-VSil_2_ **5**. The value is not decreased for POOPh-VSil_2_ **2** as the sample burns completely after first ignition.

Composite samples with comparable flame retardants are reported to achieve a classification in the literature, but they have either increased sample thickness or increased flame-retardant loadings [28,46]. While 3–4 mm-thick samples may provide a V-0 classification, investigated composite samples were only 2 mm thick. The decreased sample thickness was used in order to visualize possible effects on flame retardancy and inhibition of carbon fiber degradation.

The obtained results are as expected, since flame retardants with a mode of action in the condensed phase (e.g., PO(OPh)_2_-VSil **1**), in general, lead to worse UL-94 classifications for CFRP than flame retardants (partially) acting in the gaseous phase (e.g., DOPO-VSil **4**).

### 2.3. Characterization of Residual Fibers by SEM

The composite samples consist of the bare fibers after combustion in cone calorimetry, since the epoxy matrix degrades mostly. Carbon fibers in the RTM 6 composite 0 without flame retardants are thermally degraded during irradiation (Figure 3a). At first, single holes are generated on the surface. These grow and tunnel through the fiber with longer irradiation. Due to this, fibers break and result in lower fiber length. The mean fiber diameter is reduced to 4.0 µm simultaneously [29].

If phosphasilazanes are incorporated into the matrix, the fibers are less decomposed (Figure 3b–f). Neither tunneling holes nor broken fibers occur. Still, fiber damage is observable for samples containing phosphasilazanes. The fiber surface is not smooth, but depressions along the fiber axis appear. There are depositions in the form of agglomerated particles inside the depressions. They are also aligned along the fiber axis for the samples containing POOPh-VSil_2_ 2, SCP-VSil **3**, DOPO-VSil **4**, and Spiro-VSil_2_ **5**. Hence, these particles are not dispersed continuously. In comparison, PO(OPh)_2_-VSil **1** shows deposits in the form of connected material instead of particles. This material covers the whole fiber, lowering the number of depressions.

The residual fibers were also examined via optical microscopy in order to determine average diameters of at least 30 detected fibers (Table 3). Additionally, the minimum fiber diameter is given to show whether single respirable fibers are generated. Airborne fibers are not detectable this way. Hence, a model is necessary for evaluating the formation of respirable fiber fragments. The fiber diameter of 3 µm is here described as the lower limit defined by the WHO. Fibers below this WHO limit are present for the pure RTM 6 composite **0**, indicating respirable fiber fragments in the gaseous phase during combustion. Therefore, a reduction in fibers below the WHO limit is associated with the potential for inhibition of the generation of respirable fiber fragments. The untreated fibers have a mean diameter of 7.3 µm.

All flame-retarded samples exhibit less decreased fiber diameters compared to the pure RTM 6 composite **0** (4.0 µm). PO(OPh)_2_-VSil **1** is the sample providing the most effective combination of silazane and phosphorus moiety regarding fiber protection. The mean diameter is 6.5 ± 0.3 µm. This effect can be explained by the mode of action in the condensed phase for the phosphasilazane, which leads to a closed protective layer and a classification in UL94 burning tests in the neat resin [34].

The mean fiber diameters of the phosphasilazanes **2-5** are slightly lower compared to PO(OPh)_2_-VSil **1**, and similar within standard deviation. Still, the mean fiber diameters are also above the WHO limit of 3 µm for respirable fibers. The same result is observed for minimum fiber diameters, indicating a reduction in respirable fibers. PO(OPh)_2_-VSil **1** has the highest (5.8 µm), whilst it is lowest for Spiro-VSil_2_ **5** (4.0 µm). The difference between these two samples is 1.8 µm, indicating different fiber protection efficiency.

All examined phosphasilazanes act as inhibitors for fiber degradation. Particularly, PO(OPh)_2_-VSil **1** leads to fiber diameters beyond similar phosphorus-containing flame retardants acting in the condensed phase known from previous works (e.g., Novolac-SCP (mean fiber diameter 6.0 ± 0.7 µm), TAHHT-DDPO (6.3 ± 0.4 µm)) [11,30,47]. In order to determine the influence of the silazane moiety on fiber-degradation-inhibiting properties, the residues are further analyzed by EDX and IR spectroscopy.

### 2.4. Characterization of Residues by EDX

The average elemental compositions of the deposits on the fibers are compared from multiple EDX measurements. The carbon content is detected with high concentrations since the background of these residues is carbon fibers. Hence, the values have to be compared relatively (Table 4).

The residues from RTM 6 without a flame retardant on carbon fibers show high carbon and nitrogen contents. Both contents decrease for phosphasilazane-containing samples because the deposits mainly consist of oxygen, silicon, and phosphorus. The nitrogen content is decreasing along with carbon content. Hence, the measured nitrogen content is dominantly attributed to the fiber.

The measured carbon and nitrogen contents are highest for the deposits of SCP-VSil **3** and DOPO-VSil **4** among the investigated phosphasilazanes. Additionally, the content of oxygen, silicon, and phosphorus is the lowest. These results, at first, indicate low amounts of deposits. Further, the ratio of oxygen to silicon is high with 4.9 for SCP-VSil **3** and 4.3 for DOPO-VSil **4**. For example, the generation of SiO_2_ species is described by a ratio of only two, as approximately observed for the other phosphasilazanes **1**, **2**, and **5** (see below). Therefore, in SCP-VSil **3** and DOPO-VSil **4,** additional phosphorus structures, such as PO_4_ [48] or P_2_O_5_ [33], are present.

A higher content of silicon and oxygen related to the phosphorus content is noticeable for the particles generated from the structures of PO(OPh)_2_-VSil **1**, POOPh-VSil_2_ **2**, and Spiro-VSil_2_ **5**. Along with the low carbon concentrations, this indicates high amounts of residues on the fiber. The sample containing PO(OPh)_2_-VSil **1** generates mainly SiO_2_ species, as indicated by the lowest oxygen:silicon ratio of roughly two and the lowest relative phosphorus concentration. The relative phosphorus concentration slightly increases for POOPh-VSil_2_ **2** and Spiro-VSil_2_ **5**. This increase correlates to a higher tendency for agglomeration of residual particles, as seen for other phosphorus flame retardants [47].

The results from EDX measurements support the analyzed fiber diameters. PO(OPh)_2_-VSil **1** acts as the best inhibitor of fiber degradation among the investigated phosphasilazanes. In the SEM/EDX images, deposits consisting of mainly SiO_2_ species are observed as connected structures on the fiber surface. Consequently, fiber diameter is the highest. A high phosphorus content in the residues increases the tendency for agglomeration. The agglomerates are not dispersed on the whole fiber, leading to less inhibition of fiber degradation, as observed by lower fiber diameters, especially for SCP-VSil **3** and DOPO-VSil **4**.

In order to comprehensively determine the mechanisms of fiber protection, the residues are analyzed by IR spectroscopy.

### 2.5. Characterization of Char by IR Spectroscopy

ATR-FTIR spectra of char residues for neat resin samples after irradiation in cone calorimeter (500 s, 35 kW m^−2^, analogue to [34]) show characteristic peaks for C=C (1560 cm^−1^ and 1220 cm^−1^), P=O (1200 cm^−1^), and Si-O (1080 cm^−1^, 450 cm^−1^) vibrations (Figure 4) [49,50]. The bands at about 970 cm^−1^ result from Si-N or P-O-P vibrations [31,49]. All spectra were normalized to the band intensity of C=C vibrations at 1600 cm^−1^. These vibrations are attributed to char and result dominantly from the combustion of the matrix resin. A second vibration is observable for the residue of the matrix RTM 6 **0** at 1250 cm^−1^, slightly overlapping possible bands for P=O signals in flame-retardant structures.

Interpretation of band intensities is limited for ATR-FTIR spectra. Nevertheless, several trends can be observed. Especially for POOPh-VSil_2_ **2**, an increased amount of SiO_x_ species is formed during combustion. This is expected as this compound contains the largest amount of silicon. The similar phosphasilazane PO(OPh)_2_-VSil **1** shows the second-highest amount of SiO_x_. There is no distinct signal for SCP-VSil **3**, DOPO-VSil **4,** and Spiro-VSil_2_ **5** at 1080 cm^−1^, indicating a low amount of SiO_x_ species.

The results augment the observations from EDX measurements. Samples with a low silicon content in EDX measurements (SCP-VSil **3**, DOPO-VSil **4**) also have a low intensity for Si-O vibrations in IR spectroscopy.

### 2.6. Decomposition of Flame Retardants by Mass Spectroscopy

For further characterization of the flame-retardant mechanism, DIP-MS measurements of the flame retardants were carried out. The samples were heated up to 350 °C, which corresponds to the first degradation step of RTM 6 resin and the phosphasilazanes [34]. In this degradation step, flame retardants act and charring begins [51,52]. For samples containing PO(OPh)_x_-modified silazanes **1-2**, DIP-MS measurements resulted in the formation of phosphoramidic acid diphenyl ester (PO(OPh)_2_-NH_2_, m/e = 249, 248, 170, 94, 77, Figure 5 and Figure 6) as the main component. SCP-containing samples **3** confirmed the formation of salicylamide (m/e = 137, 120, 92), and DOPO derivatives indicated typical signals for DOPO-H fragments (m/e = 215, 168, 47). These evolved components may act as flame retardants in the gaseous phase.

Additionally, SCP-VSil **3**, Spiro-VSil_2_ **5,** and especially DOPO-VSil **4** showed the formation of 1,1,3,3-tetramethyl-1,3-divinyldisiloxane (TMDVS, m/e = 171, 159, 143, 117, 59) as a minor component. This leads to the conclusion that, under heating, a scission of the silazane bond followed by oxidation of the silane moiety occur. Rearrangement and oxidization reactions from silazanes to corresponding siloxanes are known in the literature [53,54,55].

Interestingly, for PO(OPh)_x_-containing compounds **1**–**2,** neither TMDVS nor other silicon-containing species are detected. Therefore, residual silicon compounds remain inside the sample holder and are not detected by mass spectroscopy. The silicon compounds also remain in the residue during a burning event, as shown by an increase in SiO_x_ species in the ATR-FTIR spectra or silicon content in EDX measurements for PO(OPh)_x_-containing compounds **1**–**2** after irradiation.

Therefore, the used methods describe different phenomena. The mechanism in the condensed phase is described better by ATR-FTIR and EDX, whereas the action in the gaseous phase is observed via mass spectroscopy.

### 2.7. Interlaminar Shear Strength (ILSS)

The composite samples containing PO(OPh)_2_-VSil 1 and POOPh-VSil_2_ 2 show low interlaminar shear strengths (ILSSs) between 32 and 37% compared to the sample without a flame retardant due to an irregular surface resulting from the hand lamination process. Hence, the measurements are not comparable to the others. The samples containing SCP-VSil 3, DOPO-VSil 4, and Spiro-VSil_2_ 5 show ILSSs of around 50 N mm^−2^, which is compared to the pure RTM 6 composite between 74% and 77% (Table 5). Such a decrease is also caused by other pre-ceramic materials [26], but the investigated phosphasilazanes have the advantage of being used in bulk. Hence, they can be combined with other structures, such as additional flame retardants or cross linkers, that may decrease the ILSS to a lesser extent.

## 3. Discussion

### 3.1. Influence of Chemical Structure of the Phosphasilazane on Flame-Retardant Mode of Action

The investigated phosphasilazanes generate two major silicon-containing structures during decomposition. TMDVS is released in the gaseous phase, as determined by mass spectroscopy. It is generated by a scission of the silicon–nitrogen bond, followed by a partial oxidation of the silane moiety, occurring for the phosphasilazanes SCP-VSil **3**, DOPO-VSil **4,** and Spiro-VSil_2_ **5** (Figure 7b). Flame-retardant properties are observable in the gaseous phase (Table 6) via fragments of salicylamide for SCP-VSil **3** or DOPO-H for DOPO-VSil **4**. Still, these properties are not sufficient in the UL94 burning test and cone calorimetry, since flammable TMDVS is evolving into the gaseous phase. Consequently, for SCP-VSil **3** and DOPO-VSil **4,** high amounts of phosphorus relative to silicon are observed in the residues after irradiation. Therefore, the generated particles tend to agglomerate and the fiber-protective properties are limited.

On the other hand, SiO_x_ structures are generated in the condensed phase predominantly for the phosphasilazanes PO(OPh)_2_-VSil **1** and POOPh-VSil_2_ **2,** as shown in IR and EDX measurements. Both structures show increased flame-retardant properties in the condensed phase in cone calorimetry and PO(OPh)_2_-VSil **1** in the UL94 burning test. The inorganic SiO_x_ structures do not agglomerate and increase fiber protection. They are produced via complete oxidation of the silazane moiety. The reaction may be induced by an intramolecular rearrangement via an electrophilic attack by silicon on the electron-rich P=O functionality, driven by the high energy of the formed Si-O bond (Figure 7a) [53,54]. This isomerization reaction is favored for structures with free rotation of the substituents at the phosphorus atom, as for PO(OPh)_2_-VSil **1** and POOPh-VSil_2_ **2**. It is hindered for bulky, rigid substituents at the phosphorus atom, like for ring systems in SCP-VSil **3**, DOPO-VSil **4,** and Spiro-VSil_2_ **5**, as the ring structure does not favor electron delocalization over the N-P-O group. Hence, the scission of the silicon–nitrogen bond and generation of TMDVS are favored for these molecules **3**, **4**, and **5** (Figure 7b).

The interlaminar shear strength is decreased by all phosphasilazanes. Since this parameter is the same for SCP-VSil **3**, DOPO-VSil **4**, and Spiro-VSil_2_ **5**, the chemical environment of the phosphorus atom has a neglectable influence on this value. This is known to other phosphorus-containing flame retardants as well [29,30].

Overall, the flame-retardant mode of action in the condensed phase via SiO_x_ species and fiber-protective properties can be achieved with structures without ring strain, acting mainly in the condensed phase. A decreased amount of phosphorus in the char improves fiber-protective properties via continuous barriers on the carbon fibers. The mode of action in the gaseous phase is influenced by the organic moieties at the phosphorus atom.

### 3.2. Comparison of Mode of Action in Neat Resin and in Composite

The various structures of the investigated phosphasilazanes influence flame retardancy in both neat epoxy resin samples and composites thereof, but slightly differently. Resin samples are homogeneous, containing the solved phosphasilazanes [34]. The additional carbon fibers in the composites mainly interfere physically in the burning process. They dilute the combustible material, can act as barriers, especially during delamination processes, conduct heat into the bulk material, and may increase the surface for better oxygen access [56].

It is found that for the composite samples, the investigated phosphasilazanes tend more to a mode of action in the condensed phase compared to the neat resin samples. This is because, for example, transport processes of volatiles such as gas-phase-active flame retardants are hindered by the carbon fiber plies. The flame-retardant properties of PO(OPh)_2_-VSil **1**, which acts via charring in neat resin, do not significantly change in the composite (Table 6). However, since the condensed phase mechanism is more pronounced in the composite, phosphasilazanes that accelerate burning in neat resin (POOPh-VSil_2_ **2**, DOPO-VSil **4**) gain flame-retardant properties in the composites. Even if DOPO-VSil **4** additionally acts in the gaseous phase via phosphorus moieties (DOPO-H), it is still not as efficient as PO(OPh)_2_-VSil **1**, as gas-phase-active phosphasilazane flame retardants are less effective in the composite.

SCP-VSil **3** shows greatly increased flame-retardant performance via intumescence in neat resin samples. However, SCP-VSil **3** is not as efficient in the composite, since carbon fiber plies negatively affect the generation of intumescent barriers. Even if flame-retardant salicylamide (as well as flammable TMDVS) evolves into the gaseous phase, no influence on the flame-retardant performance occurs compared to the composite without phosphasilazanes.

The aromatic moieties of phosphasilazanes in **1** to **4** typically support the flame-retardant performance, especially by a mode of action in the condensed phase. On the contrary, a structure with a cyclo-aliphatic backbone, such as Spiro-VSil_2_ **5,** does not increase the flame-retardant performance, neither in neat epoxy resin nor in the composite. Apart from that, the mode of action in the condensed phase may also be deteriorated.

## 4. Materials and Methods

### 4.1. Materials

All materials were obtained from commercial sources: High-performance epoxy resin RTM 6 and woven carbon fiber fabric G0939 (both Hexcel^®^ Corp; Stamford, CT, USA) were obtained from Lange + Ritter GmbH. (Gerlingen, Germany).

In this work, the influence of the novel phosphasilazane flame retardants from our previous work [34] on fiber protection and flame retardancy in carbon fiber composites is discussed. All molecules **1**–**5** (Figure 8 share the active center of a phosphasilazane. The examined phosphasilazanes are synthesized via metathesis reaction according to [34,57,58]: PO(OPh)_2_-VSil (*N*-Dimethylvinylsilylamidophosphorus diphenylester, **1**), POOPh-VSil_2_ (*N,N′*-bis(dimethylvinylsilyl)diamidophosphorus phenylester, **2**), SCP-VSil (2-*N*-dimethylvinylsilylamido-4*H*-1,3,2-benzodioxaphosphorine-4-one-2-oxide, **3**), DOPO-VSil (10-*N*-dimethylvinylsilylamido-9-hydro-9-oxa-10-phosphaphenanthrene-10-oxide, **4**) and Spiro-VSil_2_ (3,9-bis(dimethylvinylsilyl)diamido-2,4,8,10-tetraoxa-3,9-diphosphaspiro [5.5]undecane-3,9-dioxide, **5**).

### 4.2. Methods

Vertical UL94 burning tests (Dr.-Ing. Georg Wazau Mess- + Prüfsysteme GmbH, Berlin, Germany) were carried out analogous to DIN EN 60695-11-10 [59]. Samples with dimensions 65 mm × 13 mm × 2 mm were treated twice with a 50 kW Bunsen burner flame at a distance of 10 mm in a 20° angle for 10 s.

Cone calorimetry tests (Cone Calorimeter Pro, Dr.-Ing. Georg Wazau Mess- + Prüfsysteme GmbH, Berlin, Germany) were performed according to DIN EN ISO 5660-1 [60]. Samples were irradiated for 300–1200 s at 60 kW m^−2^ since fiber degradation is not observed for lower heat fluxes, as described in the literature [11,61]. Data were analyzed automatically using the program Cone Cal by the same supplier. The following parameters are compared after 300 s. The abbreviations of parameters are as follows: TTI = time to ignition, TTF = time to flameout, pHRR = peak heat release rate, THR = total heat released, MARHE = maximum of average rate of heat emitted, TSR = total smoke released, m_R_ = residual mass, and X = resin content. Samples were cut out of the residues with a scalpel for further analysis.

Confocal microscopy (Olympus BX50, Olympus Europa SE & Co KG, Hamburg, Germany) was carried out at 50x magnification. Scanning electron microscopy/energy-dispersive X-ray spectroscopy (SEM-EDX, EVO HD25, Carl Zeiss Microscopy GmbH, Jena, Germany) is performed with a cathode voltage of 1 kV. The used detectors are an in-lens detector or a secondary electron multiplier.

Infrared spectra (IR, Tensor 27, Bruker, Germany) were recorded with 32 scans achieving a resolution of 2 cm^−1^ in attenuated total reflectance mode on a silicon crystal. Samples were cut from the top layer of char with a scalpel after cone calorimetry.

Mass spectroscopy (MS) was performed using a direct inlet probe (DIP) from Scientific Instruments Manufacturer GmbH (Oberhausen, Germany) in combination with Inert XL Mass Selective Detector (MSD) 5975 from Agilent Technologies Inc. (Santa Clara, CA, USA). Samples were heated from 30 °C to 350 °C with 0.3 °C s^−1^, stabilizing the temperature isothermally at 30 °C, 150 °C, 250 °C, and 350 °C for 1 min each. Resulting mass spectra were analyzed using NIST MS Search by Adaptas Solutions (Palmer, MA, USA).

Interlaminar shear strength (ILSS) was tested according to DIN EN2563 [62] in a short beam shear test with a universal testing machine Z020 by ZwickRoell GmbH & Co KG (Ulm, Germany).

### 4.3. Processing of Carbon-Fiber-Reinforced Samples

Carbon-fiber-reinforced samples were produced via hand lamination of 8 fabric plies G0939 with 0° orientation, according to the literature [63]. Epoxy resin RTM 6 was heated to 120 °C and the soluble flame retardants were stirred into the resin manually, leading to a flame-retardant load of 10 wt.%. The resulting 2 mm-thick samples were cured in a closed mold at 5 bar at 160 °C for 1 h and at 180 °C for 2 h, according to the literature [11,29]. The cured carbon-fiber-reinforced composites were cut into the dimensions for the different testing methods with a water-cooled diamond saw. The samples were dried in an air-circulated oven overnight at 80 °C. The matrix content X is calculated by the subtraction of the ratio between the mass of carbon fibers and composite sample mass after curing from 1.

## 5. Conclusions

All investigated phosphasilazanes act as inhibitors of carbon fiber degradation in burning events of composites. The fiber diameter is less decreased in a fire and above the WHO limit of 3 µm compared to the composite without a flame retardant, indicating the inhibition of the generation of respirable fiber dust. The phosphasilazanes form Si-O and P-O species that bind with char, forming a flexible residue in neat resin and particles on carbon fibers in composite samples. Additionally to prohibiting fiber degradation, phosphasilazanes and especially PO(OPh)_2_-VSil **1** exhibit flame-retardant properties. They predominantly act in the condensed phase as well as in the gaseous phase, generating different phosphorus moieties. Structures with ring systems at the phosphorus atom lead to worse flame-retardant properties. They induce scission and oxidation of the silazane moiety, decreasing the silicon content in the residue and generating the flammable structure TMDVS. The generation of TMDVS does not occur for phosphasilazanes bearing less bulky and non-rigid moieties at the phosphorus moiety. The flame-retardant properties may also be increased with a synergist formulation with other flame retardants.

The flame-retardant properties are partially comparable between neat resin and composites. However, flame-retarding mechanisms are influenced by the fiber plies in the composites. A basic approach was found for neat resin samples: O-R groups bonded to the P=O group of the phosphasilazane, favor mode of action in the condensed phase and improve flame retardancy, and are not always applicable to the corresponding composite samples. Neat resin samples showing intumescence led to worse results compared to composites due to barrier effects of the carbon fiber plies, impeding intumescence and providing a high surface area for enhanced oxygen access. On the other hand, neat resin samples containing phosphasilazanes acting weakly by combined mechanisms in condensed and gaseous phases achieved improved flame-retardant properties as composites, in which especially the flame retardant mode of action in the condensed phase is enhanced by the carbon fiber plies.

Interlaminar shear strength of the investigated composites is weakened by the contained phosphasilazanes but comparable to other pre-ceramic modified samples. Since the phosphasilazanes are incorporated into the matrix, the loading may be decreased or alternatively combined with other materials, such as cross linkers, increasing the ILSS.

Overall, phosphasilazanes show high potential as a class of flame retardants that also hinder fiber degradation of carbon-fiber-reinforced polymers during burning. Their efficiency may be improved through investigation of newly synthesized structural variants, or via polymerization of the vinyl group. Additionally, their concentration may be varied or they could be used in formulations as synergistic flame retardants or by concentration variation.

## Figures and Tables

**Figure 1 molecules-28-01804-f001:**
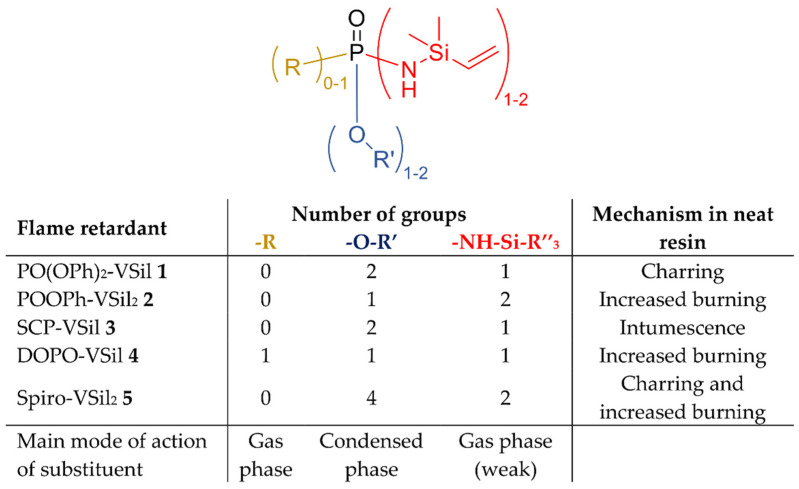
Schematic influence of structural variation in the phosphorus atom on flame-retardant properties of phosphasilazanes in neat epoxy resin. Reprinted from [34].

**Figure 2 molecules-28-01804-f002:**
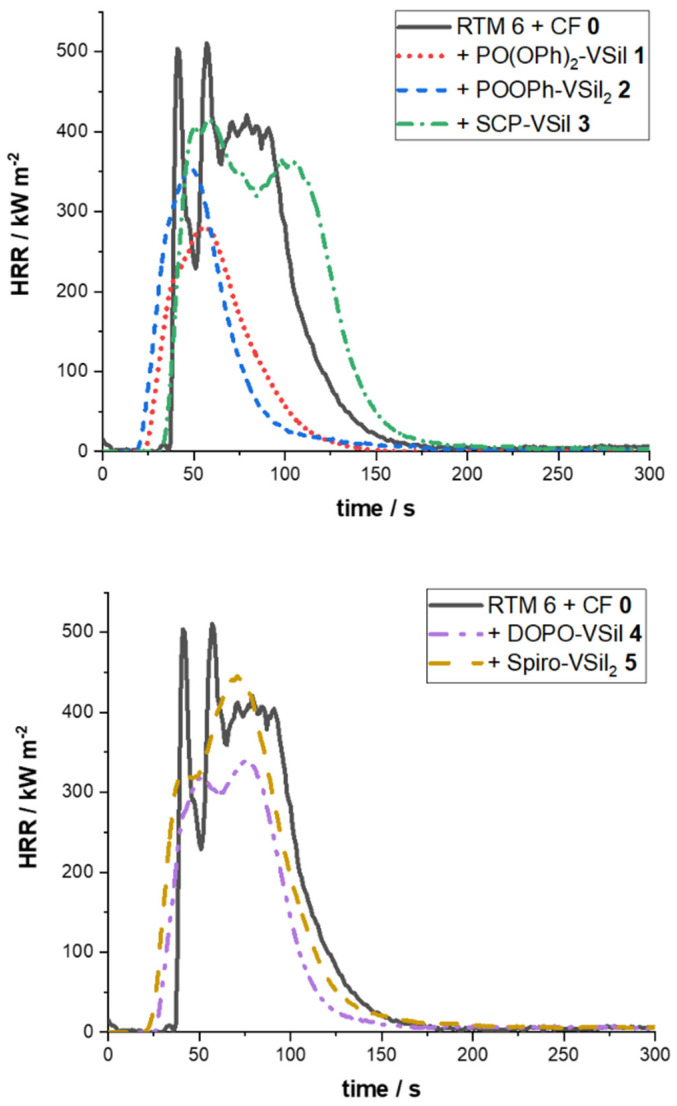
HRR curves for PO(OPh)_2_-VSil **1**, POOPh-VSil_2_ **2**, SCP-VSil **3** (**above**), and DOPO-VSil **4** and Spiro-VSil_2_ **5** (**below**) in carbon-fiber-reinforced RTM 6 composites from cone calorimetry (60 kW m^−2^).

**Figure 3 molecules-28-01804-f003:**
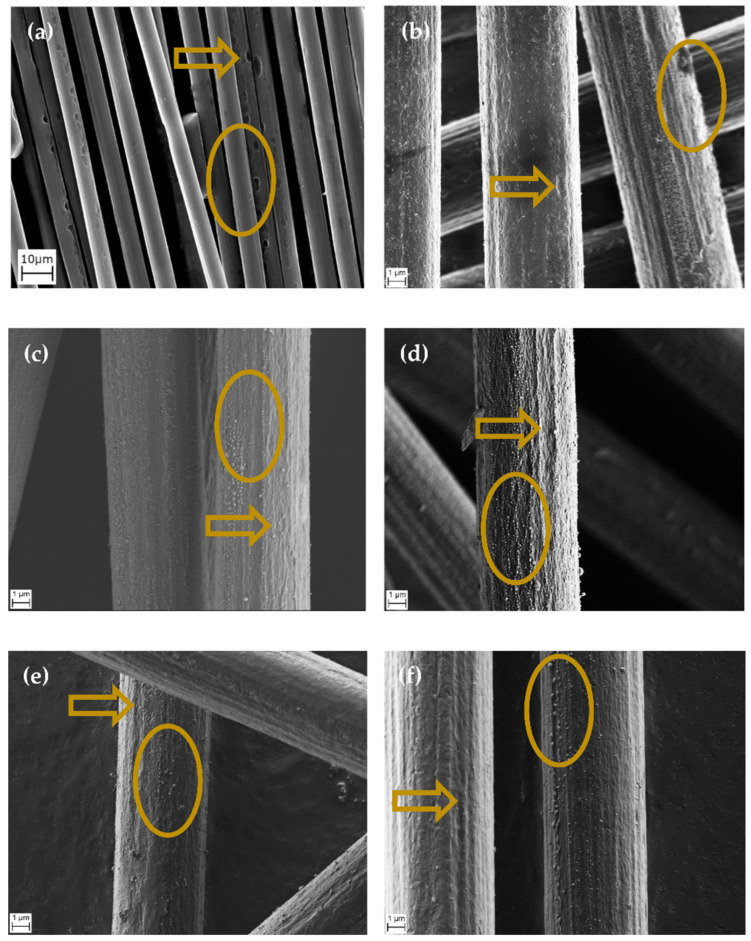
SEM images of carbon fibers after cone calorimetry (1200 s, 60 kW m^−2^). (**a**) Without flame retardant and with (**b**) PO(OPh)2-VSil **1**, (**c**) POOPh-VSil2 **2**, (**d**) SCP-VSil **3**, (**e**) DOPO-VSil **4**, (**f**) Spiro-VSil2 **5**. Exemplary defects of fibers are marked with an arrow. Areas analyzed by EDX are marked with an oval.

**Figure 4 molecules-28-01804-f004:**
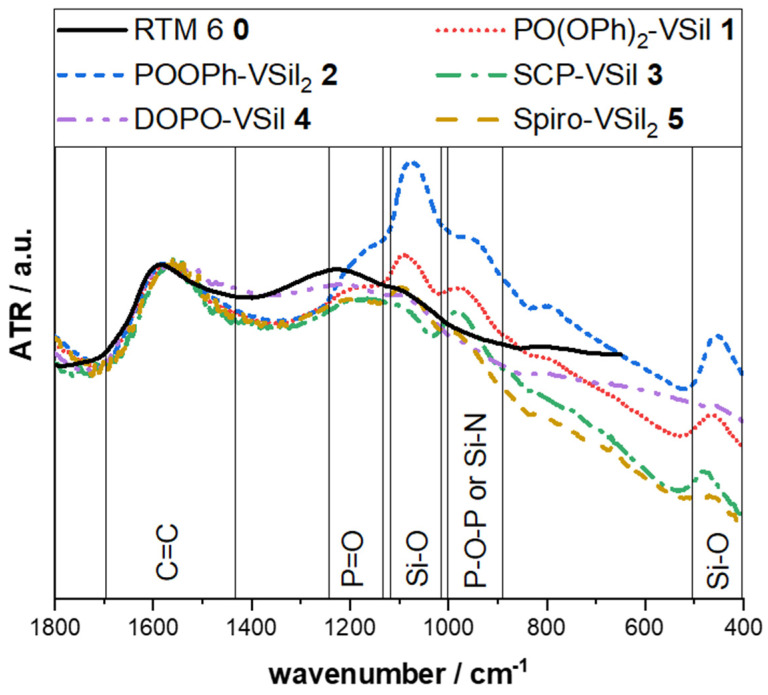
IR spectra of residues of phosphasilazanes in RTM 6 matrix after irradiation in cone calorimeter (500 s, 35 kW m^−2^).

**Figure 5 molecules-28-01804-f005:**
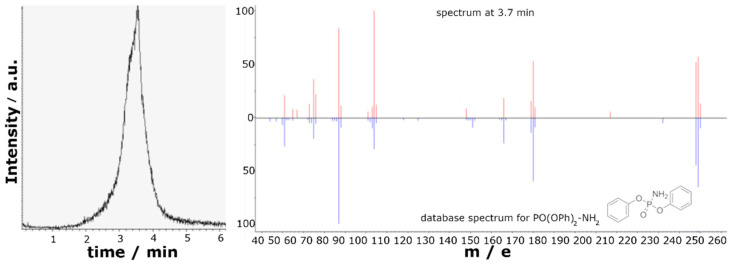
Exemplary total ion current (**left**) of PO(OPh)_2_-VSil **1** and DIP-MS spectrum at 3.7 min (**right**) of phosphoramidic acid diphenyl ester (red) with reference spectrum from database (blue).

**Figure 6 molecules-28-01804-f006:**
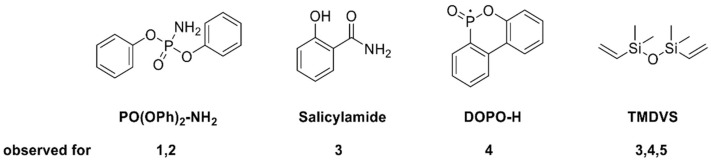
Degradation structures from phosphasilazanes in DIP-MS.

**Figure 7 molecules-28-01804-f007:**
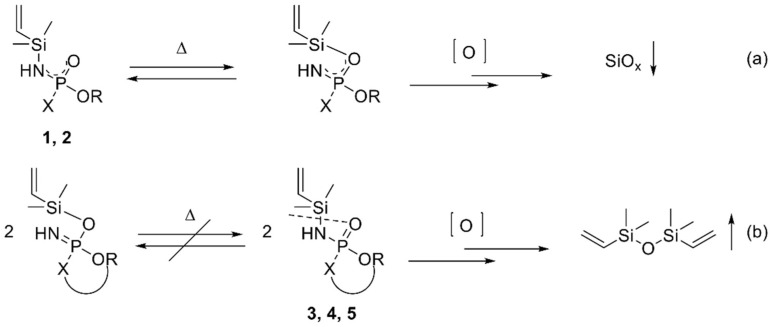
Proposed schematic decomposition pathways for phosphasilazanes without (**a**) and with (**b**) ring structure.

**Figure 8 molecules-28-01804-f008:**
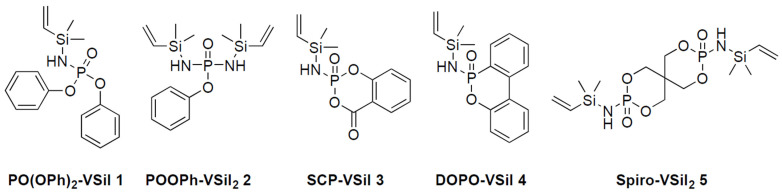
Structures and acronyms of examined phosphasilazane flame retardants. Reprinted from [34].

**Table 1 molecules-28-01804-t001:** Overview on results from cone calorimetry after 300 s at 60 kW m^−2^.

	TTI/ s	TTF/ s	pHRR/ kW m^−2^	THR X^−1^/ MJ m^−2^	MARHE/ kW m^−2^	TSR X^−1^/ m^2^ m^−2^	m_R_/ %	X
RTM 6 + CF **0**	31 ±2	193 ±42	492 ±19	72 ±1	484 ±10	3760 ±170	59 ±1	0.43
+PO(OPh)_2_-VSil **1**	18 ±4	173 ±68	260 ±32	55 ±7	271 ±8	2290 ±270	65 ±9	0.27
+POOPh-VSil_2_ **2**	19 ±1	186 ±48	342 ±18	57 ±3	431 ±37	2130 ±140	64 ±7	0.27
+SCP-VSil **3**	32 ±3	211 ±29	423 ±9	72 ±7	413 ±38	3820 ±470	48 ±7	0.49
+DOPO-VSil **4**	26 ±2	212 ±41	363 ±34	64 ±3	344 ±16	3170 ±420	51 ±7	0.39
+Spiro-VSil_2_ **5**	18 ±4	121 ±85	434 ±44	68 ±3	444 ±19	2520 ±140	52 ±5	0.45

**Table 2 molecules-28-01804-t002:** Overview of results from vertical UL94 burning test including mean first burning time (t_1_).

	Mean t_1_ in UL94 Burning Test/s	UL94-V Classification	UL94-V Classification of Neat Resin Samples [34]
RTM 6 + CF **0**	complete burning	n.c.	n.c.
+PO(OPh)_2_-VSil **1**	44	n.c.	V-1
+POOPh-VSil_2_ **2**	complete burning	n.c.	n.c.
+SCP-VSil **3**	0	n.c.	V-0
+DOPO-VSil **4**	7	n.c.	n.c.
+Spiro-VSil_2_ **5**	0	n.c.	n.c.

**Table 3 molecules-28-01804-t003:** Overview of fiber diameters of at least 30 fibers after irradiation in a cone calorimeter (60 kW m^−2^, 1200 s).

	Fiber Diameter/µm	
	Mean	Minimum
RTM 6 + CF **0**	4.0 ± 0.5	2.9
+PO(OPh)_2_-VSil **1**	6.5 ± 0.3	5.8
+POOPh-VSil_2_ **2**	5.9 ± 0.6	4.8
+SCP-VSil **3**	5.6 ± 0.5	4.4
+DOPO-VSil **4**	5.7 ± 0.5	4.7
+Spiro-VSil_2_ **5**	5.7 ± 0.7	4.0

**Table 4 molecules-28-01804-t004:** Mean elemental concentrations in deposits on carbon fibers after irradiation in cone calorimetry (1200 s, 60 kW m^−2^).

	Carbon/%	Nitrogen/%	Oxygen/%	Silicon/%	Phosphorus/%
RTM 6 + CF **0**	95.3	3.6	1.1	not detected	not detected
+PO(OPh)_2_-VSil **1**	49.7	0.9	30.6	17.3	1.5
+POOPh-VSil_2_ **2**	37.3	1.0	40.9	18.8	2.0
+SCP-VSil **3**	87.1	1.6	7.9	1.6	1.8
+DOPO-VSil **4**	88.8	1.7	6.5	1.5	1.5
+Spiro-VSil_2_ **5**	40.9	0.8	38.5	17.1	2.6

**Table 5 molecules-28-01804-t005:** Relative interlaminar shear strength (ILSS_rel_) correlated with pure RTM6 + CF.

	ILSS/N mm^−2^	ILSS_rel_/%
RTM 6 + CF	66.8 ± 4.4	100
+PO(OPh)_2_-VSil **1** ^(a)^	(25.0 ± 5.4)	(37)
+POOPh-VSil_2_ **2** ^(a)^	(21.2 ± 8.2)	(32)
+SCP-VSil **3**	51.2 ± 4.6	77
+DOPO-VSil **4**	51.2 ± 3.3	77
+Spiro-VSil_2_ **5**	49.4 ± 2.7	74

^(a)^ Samples have an irregular surface and are not comparable.

**Table 6 molecules-28-01804-t006:** Comparison of mode of action and flame-retardancy performance of phosphasilazanes from cone calorimetry for neat resin samples [34] and carbon fiber composites compared to samples without flame retardants. - = deteriorated, o = no influence, + = increased, ++ = greatly increased.

	Neat Resin		Composite	
	Main Mode of Action	Flame Retardant Performance	Main Mode of Action	Flame Retardant Performance
PO(OPh)_2_-VSil **1**	Charring	++	Condensed phase	++
POOPh-VSil_2_ **2**	Increased burning	-	Condensed phase	+
SCP-VSil **3**	Intumescence	++	Condensed and gaseous phase	o
DOPO-VSil **4**	Increased burning	-	Condensed and gaseous phase	+
Spiro-VSil_2_ **5**	ambiguous	o	ambiguous	o

## Data Availability

The raw data required to reproduce these findings are available from the corresponding author. The processed data required to reproduce these findings are available from the corresponding author.

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
