# Peer review of "Phosphasilazanes as Inhibitors for Respirable Fiber Fragments Formed during Burning of Carbon-Fiber-Reinforced Epoxy Resins"

_molecules, 2023, doi:10.3390/molecules28041804_

Round 1

Reviewer 1 Report

1. This work takes the reduction of flexible fiber fragments formed during burning of carbon fiber reinforced epoxy resins as the starting point. However, no research has been conducted on the reduction respirable fiber fragments, nor can the reduction of flexible fiber fragments be confirmed through simple experiments in the article.

2. “The silicon moieties form a polar glass that cannot bind to nonpolar char or carbon fibers, but to glass fibers. On the other hand, the phosphorus moiety is able to form phosphates that bind covalently to the char”, this view needs literature support.

3. The particles produced by the high-temperature degradation of carbon fibers have an impact on the respiratory tract, but is this impact the same as the carbon particles produced by polymer materials? So is it meaningful to study carbon fiber separately? What is the difference between the author's research and the traditional flame retardants in catalytic combustion of polymer materials?

4. “Whilst silicones and siloxanes show good in situ flame retardant properties in carbon fiber-reinforced composites, they promote fiber degradation.” Is this statement contradictory?

5. “A higher amount of silazane groups increased flammability”. In this case, why do you choose flame retardant with silazane structures?

6. “On the other hand, structures with increased silazane-content (POOPh-VSil2 2 and Spiro-VSil2 5) or with predominant mode of action in the gaseous phase like DOPO-VSil 4 provide less flame retardancy”. This is contrary to the statement in the last sentence of the abstract that N, N'- bis (dimethylylsilyl) diamidophosphorus phenylester can bring better flame retardant effect.

7. What is the role of carbon carbon double bond in flame retardant? Does it affect the curing behavior, crosslinking degree and mechanical behavior of epoxy? Does it affect the charring behavior?

8. Figure 4 Is this comparison reasonable? Is it reasonable to calculate the average diameter of only 30 fibers?

9. The SEM test method has a great relationship with the selection and preparation of samples and the operation in the SEM test process. Therefore, I do not think that the improvement of the charring performance can be verified through SEM alone? It is also not possible to prove that the number of respirable fiber fragments decreases through SEM.

10. How to verify the reduction of respirable fiber fragments through other research methods? The author must provide reasonable experimental basis to confirm the author's research purpose.

Author Response

Author’s Reply to the Review Report (Reviewer 1):

Thank you for your recommendations. It helps us to improve the publication.

Please find our comments to your recommendations below:

  1. This work takes the reduction of flexible fiber fragments formed during burning of carbon fiber reinforced epoxy resins as the starting point. However, no research has been conducted on the reduction respirable fiber fragments, nor can the reduction of flexible fiber fragments be confirmed through simple experiments in the article.

In this article, the fiber diameter of residual fibers and their fragments are examined as a model for formation of respirable fiber fragments which is state of the art (see references 1 below). This model can be seen as an indication for the formation of respirable fiber fragments, since thin fibers (< 3 µm) are necessary and thin fibers easily break. Measurements of airborne fiber fragments are complex (see references 2 below). Therefore, we used the common model for this publication. We are currently investigating this topic in the context of another research.

We added an explanation in the results and adapted the formulation in the conclusion:

“Airborne fibers are not detectable this way. Hence, a model is necessary for evaluating the formation of respirable fiber fragments. The fiber diameter of 3 µm is here described as the lower limit defined by the WHO. Fibers below this WHO limit are present for the pure RTM 6 composite 0 indicating respirable fiber fragments in the gaseous phase during combustion. Therefore, a reduction of fibers below the WHO limit is associated with the potential for inhibition of the generation of respirable fiber fragments.”

“The fiber diameter is less decreased in a fire and above the WHO limit of 3 µm compared to the composite without a flame retardant, indicating the inhibition of the generation of respirable fiber dust.”

References 1: Eibl, S. Potential for the formation of respirable fibers in carbon fiber reinforced plastic materials after combustion. Fire Mater. 2017, 41, 808–816, doi:10.1002/fam.2423.
Greiner, L.; Döring, M.; Eibl, S. Prevention of the formation of respirable fibers in carbon fiber reinforced epoxy resins during combustion by phosphorus or silicon containing flame retardants. Polymer Degradation and Stability 2021, 185, 109497, doi:10.1016/j.polymdegradstab.2021.109497.

References 2: Eibl, S. Besondere Gesundheitsgefährdung durch CFK im Brandfall. Lightweight Des 2015, 8, 26–29, doi:10.1007/s35725-014-1007-4.
T. Hertzberg, P. Blomqvist, Particles from fires - a screening of common materials found in buildings, Fire Mater. 27 (2003) 295–314. https://doi.org/10.1002/fam.837.

  1. “The silicon moieties form a polar glass that cannot bind to nonpolar char or carbon fibers, but to glass fibers. On the other hand, the phosphorus moiety is able to form phosphates that bind covalently to the char”, this view needs literature support.

Suitable references were added:

Zhang, W.; Li, X.; Fan, H.; Yang, R. Study on mechanism of phosphorus–silicon synergistic flame retardancy on epoxy resins. Polymer Degradation and Stability 2012, 97, 2241–2248, doi:10.1016/j.polymdegradstab.2012.08.002.
Zhang, P.; Fan, H.; Tian, S.; Chen, Y.; Yan, J. Synergistic effect of phosphorus–nitrogen and silicon-containing chain extenders on the mechanical properties, flame retardancy and thermal degradation behavior of waterborne polyurethane. RSC Adv. 2016, 6, 72409–72422, doi:10.1039/C6RA15869B.
Wang, X.; Hu, Y.; Song, L.; Xing, W.; Lu, H. Thermal degradation behaviors of epoxy resin/POSS hybrids and phosphorus–silicon synergism of flame retardancy. J. Polym. Sci. B Polym. Phys. 2010, 48, 693–705, doi:10.1002/polb.21939.

  1. The particles produced by the high-temperature degradation of carbon fibers have an impact on the respiratory tract, but is this impact the same as the carbon particles produced by polymer materials? So is it meaningful to study carbon fiber separately? What is the difference between the author's research and the traditional flame retardants in catalytic combustion of polymer materials?

Respirable carbon fiber fragments are generated from the carbon fibers within a polymer composite in a similar, but not identical way like from neat fibers. We showed this in a field test (see reference). Since the burning conditions change for the matrix, it is necessary to analyze the whole composite, as we did in our work. Studies from other researchers focus either on higher temperature (like burning events in engines for aviation) or on carbon fibers without polymer matrix. In traditional studies on flame retardants in catalytic combustion of polymer materials, the effect of fiber degradation and formation of respirable fiber fragments is neglected as an additional fire hazard.

Reference: Eibl, S. Besondere Gesundheitsgefährdung durch CFK im Brandfall. Lightweight Des 2015, 8, 26–29, doi:10.1007/s35725-014-1007-4

  1. “Whilst silicones and siloxanes show good in situ flame retardant properties in carbon fiber-reinforced composites, they promote fiber degradation.” Is this statement contradictory?

For flame retardancy, the effect of a substance on fiber degradation is typically not relevant. These two effects may even act in opposite directions (increasing one parameter and decreasing another) as for silicones and siloxanes. The phrase was improved to clarify this:

“Silicones and siloxanes show good in situ flame retardant properties regarding the matrix in carbon fiber-reinforced composites. [27,28] They also result in negative effects, especially in promoting fiber degradation.“

  1. “A higher amount of silazane groups increased flammability”. In this case, why do you choose flame retardant with silazane structures?

Flame retardants do not show linear correlation of performance with concentration. Above a maximum, their flame retardant effect decreases or even worsens the flammability. In comparison between different structures, the molecules containing two silazane moieties increases flammability, whilst structures with one silazane moiety improve flame retardancy. An explanatory phrase was added:

“On the other hand, a higher amount of silazane groups within one molecule at the same concentration increased flammability”

Overall, phosphorus containing silazane structures do not always increase flammability, but show flame retardancy especially in the condensed phase.

  1. “On the other hand, structures with increased silazane-content (POOPh-VSil2 2 and Spiro-VSil2 5) or with predominant mode of action in the gaseous phase like DOPO-VSil 4 provide less flame retardancy”. This is contrary to the statement in the last sentence of the abstract that N, N'- bis (dimethylylsilyl) diamidophosphorus phenylester can bring better flame retardant effect.

The IUPAC name was mistaken in the abstract. The structure PO(OPh)2‑VSil shows the best combination of flame retardancy and fiber protection. The structural name in the abstract was changed to N‑dimethylvinylsilyl-amidophosphorus diphenylester.

  1. What is the role of carbon carbon double bond in flame retardant? Does it affect the curing behavior, crosslinking degree and mechanical behavior of epoxy? Does it affect the charring behavior?

The vinyl moiety at the silazane decreases the decomposition temperature of the flame retardant in comparison to an analogue methyl moiety. This is known to literature for similar structures (references see below). In unpublished experiments, we saw the same effect. This leads to the methyl-phosphasilazane degrading after the matrix and thus to less effective flame retardant properties than in the presented vinyl-phosphasilazanes. We could not see an effect on mechanical behavior. That is why, we do not assume a crosslinking from this double bond. The following sentences were added in the section explaining our previous results:

“The vinyl moiety does not affect the mechanical properties of the matrix. It is degraded at lower temperatures than a methyl moiety. This leads to the methyl-moiety degrading after the matrix and thus to less effective flame retardant properties than in the presented vinyl-moieties.”

References: S. Hamdani, C. Longuet, D. Perrin, J.-M. Lopez-cuesta, F. Ganachaud, Flame retardancy of silicone-based materials, Polymer Degradation and Stability 94 (2009) 465–495. https://doi.org/10.1016/j.polymdegradstab.2008.11.019.
J.D. Jovanovic, M.N. Govedarica, P.R. Dvornic, I.G. Popovic, The thermogravimetric analysis of some polysiloxanes, Polymer Degradation and Stability 61 (1998) 87–93. https://doi.org/10.1016/S0141-3910(97)00135-3.

  1. Figure 4 Is this comparison reasonable? Is it reasonable to calculate the average diameter of only 30 fibers?

For other studies, we analyzed 30-50 carbon fibers each after combustion from different specimens of the same sample. Overall, more than 200 fibers were analyzed resulting in the same average fiber diameter. An increased number of fibers neither has an influence on deviation. The wording was corrected to “at least 30 fibers”.

  1. The SEM test method has a great relationship with the selection and preparation of samples and the operation in the SEM test process. Therefore, I do not think that the improvement of the charring performance can be verified through SEM alone? It is also not possible to prove that the number of respirable fiber fragments decreases through SEM.

We examined the residual fibers at different magnitude in SEM. The chosen sections are representative. The carbon fibers were prepared the same way by the same operator.

The improvement of charring is not only verified by SEM, but also by the residual mass in cone calorimetry as well as the FTIR spectroscopy.

The number of fibers with a diameter below 3 µm was decreased from 3 of 30 for RTM 6 composite 0 to 0 of 30 for all phosphasilazanes. We think, this significantly indicates, that the potential for generation of respirable fiber fragments is decreased.

  1. How to verify the reduction of respirable fiber fragments through other research methods? The author must provide reasonable experimental basis to confirm the author's research purpose.

Further research for direct proof for inhibition of formation of respirable carbon fiber fragments would be possible with a large scale fire test. Carbon fibers can be collected and analyzed via SEM. Unfortunately, this experimental setup is very complex. In a different approach (see reference), it is possible to filter and compress a part of the generated gases and soot. The filter cake is analyzed via SEM. Still, this method can only observe a part of the airborne fibers and hence is no complete proof if no respirable fiber fragments are detected. Additionally, both experimental setups are dependent from the same measurement method (SEM).

Reference: T. Hertzberg, P. Blomqvist, Particles from fires - a screening of common materials found in buildings, Fire Mater. 27 (2003) 295–314. https://doi.org/10.1002/fam.837.

Reviewer 2 Report

1. Do not use illustrations in the Introduction section.

2. It is recommended to supplement the smoke density test data.

3. It is recommended to supplement the microscopic appearance information of the sample after combustion.

4. It is suggested to reduce the number of references.

5. It is suggested to increase the discussion on shear strength in the discussion section.

Author Response

Thank you for your recommendations. It helps us to improve the publication.

Please find our comments to your recommendations below:

  1. Do not use illustrations in the Introduction section.

The illustration section was moved from the Introduction to the Result Section. The corresponding paragraphs were adjusted for a better flow of reading. The following paragraph was moved, too:

“In this publication, we showed that the chemical environment of the phosphorus atom in the phosphasilazane affects fire properties in neat epoxy resin. Especially, phosphasilazanes with an oxygen rich environment at the phosphorus atom leading to a mode of action in the condensed phase showed improved flame retardancy in neat resin. For PO(OPh)2‑VSil 1 charring occurs. SCP‑VSil 3 also exhibits intumescing char. Both structures contain two OR‑moieties at the phosphorus atom. On the other hand, structures with increased silazane-content (POOPh‑VSil2 2 and Spiro‑VSil2 5) or with predominant mode of action in the gaseous phase like DOPO‑VSil 4 provide less flame retardancy due to a carbon or nitrogen rich environment at the phosphorus atom.“

  1. It is recommended to supplement the smoke density test data.

The total smoke release per mass loss is used as a parameter to describe the factor smoke. The influence of the phosphasilazanes on this parameter are very small. Hence, the data of smoke density do not show a benefit for the study and are not included.

  1. It is recommended to supplement the microscopic appearance information of the sample after combustion.

The epoxy matrix degrades during cone calorimetry testing. Hence, the shown SEM images are equivalent to the microscopic appearance of the whole fiber. We added a sentence in the chapter Characterization of residual fibers by SEM to clarify this:

“The composite samples consist of the bare fibers after combustion in cone calorimetry, since the epoxy matrix degrades.”

  1. It is suggested to reduce the number of references.

The number of references in paragraphs with multiple references were reduced. This way we decreased the number of references by nearly 30%.

  1. It is suggested to increase the discussion on shear strength in the discussion section.

A paragraph discussing the ILSS was added in the discussion for the influence of the chemical structure:

“The interlaminar shear strength is decreased by all phosphasilazanes. Since this pa-rameter is the same for SCP-VSil 3, DOPO-VSil 4 and Spiro-VSil2 5, the chemical environment of the phosphorus atom has a neglectable influence on this value. This is known to other phosphorus containing flame retardants as well.“

Round 2

Reviewer 1 Report

There is no comments.